# Do Gender, Age, Body Mass and Height Influence Eye Biometrical Properties in Young Adults? A Pilot Study

**DOI:** 10.3390/ijerph182111719

**Published:** 2021-11-08

**Authors:** Štefanija Kolačko, Jurica Predović, Antonio Kokot, Damir Bosnar, Vlatka Brzović-Šarić, Borna Šarić, Slaven Balog, Kristina Milanovic, Domagoj Ivastinovic

**Affiliations:** 1University Hospital “Sveti Duh”, 10000 Zagreb, Croatia; kolackostefy@gmail.com (Š.K.); damir.bosnar@gmail.com (D.B.); brzsar@gmail.com (V.B.-Š.); bornasaric@gmail.com (B.Š.); 2Department of Nursing and Palliative Medicine, Faculty of Dental Medicine and Health, Josip Juraj Strossmayer University of Osijek, 31000 Osijek, Croatia; 3Department of Ophthalmology and Optometry, Faculty of Dental Medicine and Health, Josip Juraj Strossmayer University of Osijek, 31000 Osijek, Croatia; 4Department of Ophthalmology and Optometry, Faculty of Medicine, Josip Juraj Strossmayer University of Osijek, 31000 Osijek, Croatia; 5Department of Anatomy and Neuroscience, Faculty of medicine, University of Josip Juraj Strossmayer, 31000 Osijek, Croatia; slavenbalog@gmail.com (S.B.); krisscure@gmail.com (K.M.); 6Dr. Balog Ophthalmology Clinic, 31000 Osijek, Croatia; 7Department of Ophthalmology, Medical University of Graz, 8036 Graz, Austria; domagoj.ivastinovic@medunigraz.at

**Keywords:** biometry, cornea, optic nerve, anterior chamber, lens, crystalline

## Abstract

Background: Do gender, age, body mass and height influence eye biometrical properties in young adults? Methods: A total of 155 eyes (92 female, 63 male) of healthy subjects between the ages of 18 and 39 years were included in the study. The subjects’ gender and age were recorded, and their body mass, height and biometrical properties of the eyes were measured. Results: The male subjects had significantly thicker and flatter corneas and lower minimal rim-to-disk ratios than the female subjects did. In both genders, age showed strong, negative correlations with anterior chamber depth and pupil diameter and a positive correlation with lens thickness. We also found significant, negative correlations between body height and mass with keratometry measurements, negative correlations between body height and optic disk rim area and rim volume, and positive correlations between body mass and axial length in both genders. Conclusions: Biometric eye parameters differ among people. In addition to age and gender, which are usually taken into consideration when interpreting ocular biometry findings, we strongly suggest that body height and mass should be also routinely considered when interpreting eye biometry data, as these factors have an impact on ocular biometry.

## 1. Introduction

Eye biometry is a diagnostic method used to measure the eye’s anatomical structures. The analysis of the eye’s biometrical properties is crucial in diagnostic and therapeutic procedures in ophthalmology [1]. In particular, the eye’s biometrical properties are essential to calculate the power of the intraocular lens in cataract surgery, in refractive surgery and in the diagnostics and monitoring of patients with age-related macular degeneration, glaucoma and other conditions [2,3,4]. In the past, ultrasound contact biometry was the most widely used biometric method. Today, more precise, noncontact, easy-to-use optical biometry methods are used.

The biometrical properties of the eye are specific to each eye. Thus, these properties vary between individuals and populations. The variations in biometrical properties may be attributed to height, mass, gender, heredity and other factors [5,6]. Furthermore, the biometrical properties of the eye may change throughout an individual’s lifetime and with the physiological aging process.

The majority of previous studies on eye biometrics have been based on data collected from elderly patients (presbyopic age groups) with cataracts or other pathological eye conditions [2,3]. Thus, studies on healthy middle-aged populations are scarce. Therefore, we decided to investigate healthy pre-presbyopic young adults (18 to 39 years of age) whose refractive error is minimal (spherical components of less than −1.00 diopters (D) or less than +2.50 spherical D, and oblique astigmatism of less than 1.00 D and less than 1.50 D with or without astigmatism). To the best of our knowledge, this is the first study that has focused on the correlation between the eye biometric properties of middle-aged healthy adults with age, gender, body mass and height. Our study includes comprehensive anterior and posterior segment parameters and general biometric parameters, including age, gender, body mass and height.

In a study conducted in an elderly Chinese population, Chen et al., stated that age and gender were the most consistent predictors of ocular biometrics [7]. Nangia V. et al., conducted a study in a rural area of central India with healthy subjects aged 30 years and older and found that body height and eye size were related [8]. In a Latino population older than 40 years, Shufelt et al., reported that there were differences among respondents depending on age and gender [1]. We aimed to determine whether age, gender, body mass and height influence the eye’s biometrical properties, and whether these factors should be considered when interpreting ocular biometry analysis data. We believe that the results of this study will contribute to a better understanding of ocular morphology and improve biometry-based diagnostic and therapeutic procedures in ophthalmology.

### 1.1. Patients/Materials and Methods

This study was approved by the Ethical Committee of the University Hospital “Sveti Duh”, Zagreb, Croatia, and was conducted following the guidelines of the Helsinki Declaration between February 2018 and April 2020.

Subjects were recruited from the student and working population that underwent annual systematic ophthalmic examinations at the University Hospital “Sveti Duh”, Zagreb, Croatia. Pre-presbyopic middle-aged (18 to 39 years old) Caucasian subjects of both genders underwent complete ophthalmic examinations prior to inclusion in the study. Only subjects with a visual acuity at least 0.00 logMAR were included in the study. All subjects with a previous history of ocular trauma, ocular inflammation or other eye diseases, including strabismus, amblyopia, significant myopia and hyperopia, were excluded from the study. Only near-emmetropic subjects, with spherical components of less than −1.00 diopters (D) or less than +2.50 spherical D, and oblique astigmatism of less than 1.00 D and less than 1.50 D with or without the rule astigmatism, were included in the study. Both eyes were examined in each subject, but only right eye biometrical data were used for further analysis.

Demographic data (gender and date of birth) were recorded, and subjects’ ages were calculated. Body mass and body height were measured while the subjects were wearing light clothes without shoes.

Two optical biometry methods, optical low-coherence reflectometry and optical coherence tomography (OCT) of the macular region and optical disc, were used to measure the biometrical properties of both eyes of each subject. We ensured that subjects had not previously used any medication that would affect biometrical properties of the eye. All measurements were taken in a single day by an experienced operator under uniform mesopic conditions.

The noncontact optical low-coherence reflectometry device Lenstar LS 900^®^ (HAAG-STREIT AG, Gartenstadtstrasse 10, 3098 Koeniz, Switzerland) was used to measure the axial length (AL), central corneal thickness (CCT), anterior chamber depth (ACD), lens thickness (LT), the flattest and steepest keratometry (K1 and K2, respectively), white-to-white corneal diameter (WTW) and pupil diameter (PD). The spectral domain OCT device Canon HS-100 SD-OCT (Canon Inc., Tokyo, Japan) was used to measure the minimal macular thickness, average macular thickness and volume in the central 6 mm zone (early treatment diabetic retinopathy study (ETDRS) circle) of the macular region and optic disc. The disc area surface, rim area surface, cup volume, rim volume and peripapillary retinal nerve fiber layer (RNFL) thickness (average and inferior (RNFL-I), superior (RNFL-S), nasal (RNFL-N) and temporal (RNFL-T) quadrant) were measured in the standardized TSNIT (T-temporal, S-superior, N-nasal, I-inferior, T-temporal) pattern.

### 1.2. Statistical Analysis

To observe the mean effect of the difference of the numerical variables between the two independent groups of subjects, we used a significance level of 0.05 and a test power of 0.80 (G*Power analysis) with a minimal sample size of 128. Categorical data were represented by the absolute and relative frequencies. The normality of the numerical variable distribution was studied using the Shapiro–Wilk test. Numerical data were described by the mean and the 25% and 75% interquartile ranges. Differences between variables in two independent groups were tested by Student’s *t*-test. The correlation between the numerical variables was evaluated using Pearson’s correlation coefficient (r). All *p* values are two-sided. The significance level was set to Alpha = 0.05. The statistical analysis was performed using SPSS Statistics 17.0 (SPSS Inc., Chicago, IL, USA).

## 2. Results

The study included 155 healthy adults from Zagreb County, Croatia, totaling 92 (59.3%) females and 63 (40.7%) males. The mean age of the enrolled subjects was 22.69 years for the female subjects and 23.96 years for the male subjects. In general, the male subjects were significantly taller (181.38 to 168.34 cm) and heavier (84.09 to 67.72 kg) than the female subjects. Detailed biometric characteristics of enrolled subjects are listed in Table 1.

The male subjects had significantly thicker and flatter corneas than the female subjects. We found CCT values of 564.65 µm in males and 543.66 µm females. The flattest keratometry values were 42.25 D and 43.02 D in males and females, and the steepest keratometry values were 43.23 D and 43.84 D in males and females, respectively (Table 1). The male subjects also had lower minimal rim to disk ratios than the female subjects did, with values of 0.13 and 0.16 for males and females, respectively (Table 1). The female subjects had wider pupils and shorter eyeball axial lengths than the male subjects did (4.62 mm and 4.35 mm, *p* = 0.056 and 23.46 mm to 23.69 mm; *p* = 0.061, respectively), but these differences were not significant. There were no other significant differences in the observed eye biometry parameters between genders (Table 1).

The correlations between age, body height, body mass and eye biometry data in female and male subjects are presented in Table 2 and Table 3. In both male and female subjects, there were positive correlations between age and lens thickness (*p* = 0.000, *p* = 0.001), and there were negative correlations between anterior chamber depth (*p* = 0.002, *p* = 0.010) and pupil diameter (*p* = 0.000, *p* = 0.0.006). The flattest (K1) and steepest (K2) keratometry values (expressed in diopters), disk rim area and rim volume showed strong, negative correlations with body height and mass in both genders. Disk rim volume and area were strongly negatively correlated with body height in both genders (*p* = 0.000 for males and *p* = 0.009 for female). Body mass showed a positive correlation with eyeball axial length.

## 3. Discussion

Biometric methods have significant scientific and practical applications in modern ophthalmology [9,10]. Studies have shown that the eye’s quantitative static and dynamic measurements may explain several physiological and pathological processes [11,12,13]. Despite the rapid development and accessibility of optically based eye biometry, eye biometry data on young and middle-aged populations are still lacking. To the best of our knowledge, there are no comprehensive studies comparing anterior and posterior eye segment biometric features obtained by modern optical biometry technologies with general biometry features, such as body height and mass, in healthy young adult populations. Our study is the first with comprehensive eye and general biometry measurements in the healthy young adult population and was carried out in Croatia.

Corneal curvature and corneal thickness are important parameters in refractive surgery because these parameters are used in intraocular lens power calculation before cataract surgery and planning of corneal refractive surgery procedures. We found that, in adults younger than 40 years of age, central corneal thickness and keratometry values (both the flattest (K1) and the steepest (K2)) significantly differed in male and female subjects (males had thicker corneas and flatter corneal curvatures than females). Gender differences in keratometry may be due to the fact that males were taller than females, since body height had a negative correlation with both the flattest and the steepest corneal meridian curvatures (expressed in diopters). Central corneal thickness correlated with body height in male subjects (R = 0.337, *p* = 0.007) but not in female subjects (R = 0.005, *p* = 0.960). Therefore, gender differences in central corneal thickness cannot be fully explained by this correlation. Still, corneal thickness and keratometry differed between male and female subjects.

We found that the optic disk, rim area and rim volume showed negative correlations with body height. Therefore, we can conclude that taller people have thinner disk rims. There were no correlations between the whole disk area size and body height. Since the male subjects were taller than the female subjects, one would expect males to have thinner disk rims than females. However, we found no difference regarding gender (Table 1). Jost et al., found that taller body height was associated with a lower rate of open-angle glaucoma, which is unexpected considering that taller people have thinner disk rims [14]. We conclude that body height should be considered when analyzing optic disk rim volume in young adults, especially in glaucoma diagnostics and neuro-ophthalmology [15].

The anterior chamber becomes shallower with age as the result of physiological lens thickening, but these findings were observed in older subjects. Richdal et al., found that lens thickness increased 0.03 mm per year in an elderly population (30–50 years old) in Ohio [16]. In a study of respondents older than 40 years of age who were scheduled for cataract surgery, Bosnar et al., found that lens thickness changed with age. Moreover, lens instability depended on lens thickness, and the depth of anterior chamber decreased with age [17]. Our study is the first to describe lens thickening and anterior chamber shallowing in young adults (R = 0.576, *p* = 0.000 for males and R = 0.353, *p* = 0.000 for females).

Pupil diameter also decreased with age in both genders (R = −0.537, *p* = 0.000 for males and R = −0.208, *p* = 0.006 for females). This is consistent with previous findings [18,19].

Although not statistically significant (*p* = 0.061), we found that females had smaller axial length, shallower anterior chambers (*p* = 0.164) and wider pupils (*p* = 0.056) than males did, which might contribute to the increased incidence of primary angle closure glaucoma in females. In females, the incidence of primary angle closure glaucoma increases with age [20]. This might be due to the correlation of age with lens thickening (*p* = 0.001) and anterior chamber shallowing (*p* = 0.010), which we found in female subjects in this study. On the contrary, females had steeper corneas than men did (*p* < 0.02), which may be a protective factor for primary angle closure glaucoma, along with a pupil diameter which decreases with age (*p* = 0.006).

We found that body mass had a positive correlation with eyeball axial length (*p* = 0.000; *p* = 0.009) and a negative correlation with the steepest (*p* = 0.001; 0.032) and the flattest (*p* = 0.001; *p* = 0.010) corneal meridian curvature (expressed in diopters) both for male and female subjects.

The correlation of body height with eyeball axial length was positive and statistically significant in male but not in female subjects (*p* = 0.000; *p* = 0.086). We found a significant correlation of body height with the steepest (*p* = 0.000; *p* = 0.007) and the flattest (*p* = 0.000; *p* = 0.005) corneal meridian curvature for both male and female subjects, respectively. Nangia V. et al., studied eye biometry measured by ultrasonography in a rural area of central India in healthy subjects aged 30 years and older. The researchers found that taller subjects had larger eyes with flatter corneas [8]. Taller subjects were more likely to have larger eyes with longer axial length (+0.23 mm for each 10 cm increase in height) and deeper anterior chambers (+0.03 mm for each 10 cm increase in height). Using a more precise technique, in a Croatian population, we found that taller males, but not taller females, have longer AL, deeper ACD, flatter corneas and deeper anterior chambers. Since we found that body mass correlates well with eyeball axial length in both genders, but body height does not significantly correlate with eyeball axial length in females, we conclude that body mass is in better correlation with eyeball axial length in both genders and should be taken in consideration when interpreting eye biometry.

In a rural area of central India, Nangia V. et al., found that the mean CCT was 514 +/− 33 µm. The researchers found that CCT values were significantly associated with younger age, male gender, higher BMI, lower corneal refractive power, deeper anterior chamber, thicker lens and shorter axial length [8]. Using more precise optical methods, in a Croatian population, we found slightly higher CCT values (564.65 µm in males and 543.66 µm in females). In addition, we observed that CCT values showed positive correlation with body height in males (*p* = 0.007). The significant difference in CCT between populations could be attributed to different measurement techniques, as presented by Leung et al., (compared with ultrasound pachymetry, OCT consistently overestimated the CCT by a mean of 23 µm) [21]. Therefore, we can conclude that CCT values differ between populations and genders. These facts should be considered when performing applanation tonometry and refractive surgery [4,22,23].

Numerous studies have also investigated the effects of growth hormone and insulin on eye development, final eyeball dimensions and body height [24]. Other factors, such as diet, genetics, environmental factors and educational-level behaviors, may also have an impact on changes of the biometric characteristics of the eye [25,26]. We did not investigate these factors in our study.

Using OCT technology, we confirmed the findings of other authors that average macular thickness and volume in the central 6 mm ETDRS circle were higher in male subjects than in female subjects. However, these gender differences were not statistically significant in our case (*p* = 0.078 and *p* = 0.076, respectively) [27]. Wong KS found that retinal thickness is related to sex, age and axial length [28]. We could not confirm these correlations. However, we found a negative correlation between the average macular thickness and volume and body mass (*p* = 0.028 and *p* = 0.031, respectively) in male subjects but not in female subjects. Our results might differ from the results of Won KS due to differences in the observed populations (Won observed an older population, 55.6 +/− 16.4 years). Macular thinning, which starts in older age groups, is likely not yet present in young adults.

Patel et al., described the association of RNFL thickness loss with age between 19 and 76 years [29]. However, our data do not show a decrease in the average RNFL thickness with age between 18 and 39 years. Our results indicate, for the first time, that the decrease in RNFL thickness is not present in young adults, but it occurs later in life.

We found that age, gender, body height and body mass have a significant impact on some aspects of ocular biometry, even in young adults. We found that the anterior chamber depth shallows, the lens thickness increases and the pupil diameter decreases with age, even in young adults. Furthermore, the axial length correlates positively with body mass, while keratometry correlates negatively with body mass. Keratometry optic disk and rim volume correlate negatively with body height.

## 4. Conclusions

Although there are some limitations of this study, like relatively small sample size and not taking in consideration the refractive error (although minor), it gave us the insight into correlations of ocular biometry with age, gender, body height and body mass. The results of this study are a step forward in defining the normal biometrical properties of the eye and their dependence on gender, body height and body mass in younger populations. In addition to age and gender, which are usually taken into consideration when interpreting ocular biometry findings, we strongly suggest that body height and mass should be also routinely considered when interpreting eye biometry data, as these factors have an impact on diagnostic, prognostic and therapeutic procedures based on ocular biometry. These results may help in distinguishing normal and pathological findings and potentially lead to earlier diagnoses of various ocular pathological conditions. To achieve that goal, further, large-scale studies are needed.

## Figures and Tables

**Table 1 ijerph-18-11719-t001:** Distribution and group differences of age, body height, body mass and eye biometric parameters in female (N = 92) and male (N = 63) subjects. A Student’s *t*-test was used to compare the means of each parameter between the male and female subjects.

	Median (Interquartile Range 25–75%)		Mean Difference (95% Confidence Interval of the Difference)	*p*-Value
	Female	Male		
Age (y)	21.00 (19.00–24.00)	23.00 (19.00–27.00)	−1.27 (−2.95 to 4.09)	0.137
Body height (cm)	168.00 (164.00–172.00)	180.00 (176.00– 188.00)	−13.03 (−15.14 to 10.93)	0.000 **
Body mass (kg)	61.50 (57.00–67.00)	83.00 (74.00–95.00)	−21.36 (−24.83 to 17.90)	0.000 **
AL (mm)	21.83 (19.95–24.04)	23.62 (23.13–24.17)	−0.22 (−0.46 to 0.01)	0.061
CCT (µm)	541.00 (520.25–568.75)	566.00 (539.25–584.50)	−20.99 (−31.02 to 10.95)	0.000 **
ACD (mm)	3.03 (2.89–3.16)	3.04 (2.88–3.29)	−0.06 (−0.15 to 0.02)	0.164
LT (mm)	3.62 (3.51–3.80)	3.58 (3.46–3.67)	0.04 (−0.03 to 0.12)	0.283
K1/flat (D)	43.09 (42.03–44.05)	42.20 (41.35–43.50)	0.56 (0.10 to 1.03)	0.017 *
K2/steep (D)	43.94 (42.82–44.83)	42.87 (42.17–44.45)	0.60 (0.10 to 1.11)	0.018 *
WTW (mm)	12.31 (12.09–12.58)	12.37 (12.13–12.66)	−0.03 (−0.15 to 0.08)	0.534
PD (mm)	4.47 (3.89–5.07)	4.23 (3.74–4.71)	−0.27 (−0.00 to 0.54)	0.056
Minimal macular thickness (µm)	219.00 (212.75–226.00)	222.00 (214.00–232.00)	−2.82 (−8.11 to 2.45)	0.292
Average macular thickness in 6 mm ETDRS circle (µm)	313.00 (303.00–320.00)	315.00 (309.00–323.00)	−3.11 (−6.58 to 0.35)	0.078
Macular volume in 6 mm ETDRS circle (mm^3^)	8.84 (8.57–9.04)	8.91 (8.73–9.12)	−0.08 (−0.18 to 0.00)	0.076
Disc area (mm^2^)	1.96 (1.77–2.30)	1.97 (1.79–2.30)	−0.00 (−0.11 to 0.11)	0.098
Rim area (mm^2^)	1.61 (1.40–1.80)	1.63 (1.35–1.91)	0.01 (−0.08 to 1.11)	0.777
Cup volume (mm^3^)	0.04 (0.01–0.11)	0.06 (0.03–0.14)	0.00 (−0.03 to 0.03)	0.931
Rim volume (mm^3^)	0.35 (0.27–0.47)	0.36 (0.25–0.43)	0.01 (−0.02 to 0.05)	0.463
C/D area	0.15 (0.08–0.27)	0.18 (0.12–0.28)	−0.01 (−0.05 to 0.03)	0.598
C/D vertical	0.38 (0.27–0.50)	0.39 (0.33–0.51)	−0.02 (−0.07 to 0.02)	0.357
C/D horizontal	0.43 (0.30–0.54)	0.45 (0.34–0.56)	0.02 (−0.07 to 0.03)	0.424
R/D minimum	0.16 (0.10–0.22)	0.11 (0.09–0.15)	0.03 (0.01 to 0.06)	0.002 **
TSNIT Average (µm)	103.00 (98.00–110.00)	105.00 (98.00–108.00)	−1.35 (−5.07 to 2.36)	0.472
I (µm)	135.00 (124.50–147.00)	129.00 (120.00–139.00)	4.53 (−0.39 to 9.46)	0.071
S (µm)	123.50 (114.25–137.75)	128.00 (119.00–134.00)	−1.77 (−6.36 to 2.82)	0.447
N (µm)	80.50 (73.00–89.75)	88.00 (75.00–94.00)	−3.70 (−7.89 to 0.45)	0.083
T (µm)	74.20 (67.25–81.75)	75.00 (68.00–81.00)	−0.02 (−3.15 to 3.11)	0.988

** Correlation is significant at the 0.01 level (2-tailed); * Correlation is significant at the 0.05 level (2-tailed); D—diopters; AL—axial length; CCT—central corneal thickness; ACD—anterior chamber depth; LT—lens thickness; K1/flat—the flattest meridian corneal curvature; K2/steep—the steepest meridian corneal curvature; WTW—white-to-white corneal diameter; PD—pupil diameter; ETDRS—early treatment diabetic retinopathy study; C/D—optic disk cup to disk ratio; R/D—optic disk rim to disk ratio; TSNIT—average temporal, superior, nasal and inferior quadrant retinal nerve fiber layer thickness; I—retinal nerve fiber layer thickness in inferior quadrant; S—retinal nerve fiber layer thickness in superior quadrant; N—retinal nerve fiber layer thickness in nasal quadrant; T—retinal nerve fiber layer thickness in temporal quadrant.

**Table 2 ijerph-18-11719-t002:** The Pearson correlations of age, body height and body mass with eye biometric parameters in female (N = 92) subjects.

	Pearson Correlation (*p* Value)		
	Age/Years	Body Height/cm	Body Mass/kg
AL (mm)	−0.006 (0.955)	0.180 (0.086)	0.269 ** (0.009)
CCT (µm)	0.034 (0.749)	0.005 (0.960)	0.074 (0.484)
ACD (mm)	−0.267 ** (0.010)	−0.046 (0.661)	0.007 (0.949)
LT (mm)	0.353 ** (0.001)	0.137 (0.199)	−0.156 (0.143)
K1/flat (D)	−0.048 (0.647)	−0.291 ** (0.005)	−0.267 ** (0.010)
K2/steep (D)	−0.080 (0.446)	−0.281 **(0.007)	−0.224 * (0.032)
WTW (mm)	−0.232 * (0.026)	0.202 * (0.054)	0.026 (0.809)
PD (mm)	−0.208 ** (0.006)	−0.069 (0.512)	−0.019 (0.861)
Minimal macular thickness (µm)	0.06 (0.576)	0.158 (0.137)	−0.028 (0.795)
Average macular thickness in ETDRS circle (µm)	0.069 (0.519)	−0.004 (0.970)	−0.103 (0.334)
Macular volume in ETDRS circle (mm^3^)	0.0565 (0.545)	−0.004 (0.971)	−0.103 (0.334)
Disc area (mm^2^)	−0.044 (0.675)	−0.071 (0.502)	−0.018 (0.865)
Rim area(mm^2^)	−0.150 (0.153)	−0.215 * (0.040)	−0.033 (0.753)
Cup volume (mm^3^)	0.127 (0.226)	0.131 (0.214)	0.062 (0.560)
Rim volume (mm^3^)	−0.107 (0.312)	−0.272 ** (0.009)	−0.035 (0.743)
C/D area	0.124 (0.238)	0.139 (0.188)	0.072 (0.493)
C/D vertical	0.085 (0.422)	0.164 (0.119)	0.048 (0.653)
C/D horizontal	0.175 (0.196)	0.113 (0.282)	0.074 (0.486)
R/D minimum	−0.147 (0.165)	−0.102 (0.335)	0.035 (0.743)
TSNIT Average (µm)	0.104 (0.323)	−0.052 (0.623)	−0.048 (0.648)
I (µm)	0.020 (0.851)	−0.118 (0.262)	0.064 (0.545)
S (µm)	0.058 (0.583)	−0.195 (0.369)	−0.051 (0.627)
N (µm)	0.033 (0.754)	0.048 (0.648)	0.032 (0.764)
T (µm)	0.107 (0.312)	−0.208 * (0.047)	0.010 (0.925)

** Correlation is significant at the 0.01 level (2-tailed); * Correlation is significant at the 0.05 level (2-tailed); D—diopters; AL—axial length; CCT—central corneal thickness; ACD—anterior chamber depth; LT—lens thickness; K1/flat—the flattest meridian corneal curvature; K2/steep—the steepest meridian corneal curvature; WTW—white-to-white corneal diameter; PD—pupil diameter; ETDRS—early treatment diabetic retinopathy study; TSNIT—average temporal, superior, nasal and inferior quadrant retinal nerve fiber layer thickness; I—retinal nerve fiber layer thickness in inferior quadrant; S—retinal nerve fiber layer thickness in superior quadrant; N—retinal nerve fiber layer thickness in nasal quadrant; T—retinal nerve fiber layer thickness in temporal quadrant; C/D—cup-to-disk ratio; R/D—rim-to-disk ratio.

**Table 3 ijerph-18-11719-t003:** The correlations of age, body height and body mass with eye biometric parameters in male (N = 63 eyes) subjects.

	Pearson Correlation (*p* Value)		
	Age (Year)	Body Height (cm)	Body Mass (kg)
AL (mm)	0.156 (0.218)	0.448 ** (0.000)	0.319 ** (0.011)
CCT (µm)	0.150 (0.237)	0.337 ** (0.007)	0.269 * (0.035)
ACD (mm)	−0.391 ** (0.002)	0.377 ** (0.004)	0.038 (0.780)
LT (mm)	0.576 ** (0.000)	−0.102 (0.448)	0.090 (0.501)
K1/flat (D)	−0.072 (0.573)	−0.533 ** (0.000)	−0.420 ** (0.001)
K2/steep (D)	−0.063 (0.621)	−0.530 ** (0.000)	−0.427 ** (0.001)
WTW (mm)	0.130 (0.304)	0.207 (0.107)	0.186 (0.147)
PD (mm)	−0.537 ** (0.000)	0.109 (0.400)	−0.293 * (0.021)
Minimal macular thickness (µm)	−0.087 (0.503)	−0.170 (0.195)	−0.068 (0.607)
Average macular thickness in ETDRS circle (µm)	−0.141 (0.275)	−0.185 (0.157)	−0.283 * (0.028)
Macular volume in ETDRS circle (mm^3^)	−0.134 (0.298)	−0.185 (0.156)	−0.279 * (0.031)
Disc area (mm^2^)	0.120 (0.354)	−0.186 (0.156)	−0.018 (0.891)
Rim area (mm^2^)	−0.052 (0.691)	−0.304 * (0.020)	−0.193 (0.147)
Cup volume	0.230 (0.077)	0.219 (0.098)	0.287 ** (0.029)
Rim volume	−0.069 (0.602)	−0.460 ** (0.000)	−0.293 * (0.026)
C/D area	0.285 * (0.027)	0.328 * (0.012)	0.329 * (0.012)
C/D vertical	0.283 * (0.028)	0.362 ** (0.005)	0.330 * (0.011)
C/D horizontal	0.295 (0.249)	0.357 ** (0.006)	−0.393 ** (0.002)
R/D minimum	−0.274 * (0.034)	−0.031 (0.819)	−0.069 (0.606)
TSNIT Average (µm)	0.311 ** (0.014)	−0.072 (0.587)	0.342 ** (0.008)
I (µm)	0.179 (0.164)	−0.176 (0.178)	0.172 (0.188)
S (µm)	0.361 ** (0.004)	−0.105 (0.426)	0.185 (0.156)
N (µm)	0.177 (0.169)	0.166 (0.205)	0.438 ** (0.000)
T (µm)	0.149 (0.248)	−0.154 (0.240)	−0.009 (0.945)

** Correlation is significant at the 0.01 level (2-tailed); * Correlation is significant at the 0.05 level (2-tailed); D—diopters; AL—axial length; CCT—central corneal thickness; ACD—anterior chamber depth; LT—lens thickness; K1/flat—the flattest meridian corneal curvature; K2/steep—the steepest meridian corneal curvature; WTW—white-to-white corneal diameter; PD—pupil diameter; ETDRS—early treatment diabetic retinopathy study; TSNIT—average temporal, superior, nasal and inferior quadrant retinal nerve fiber layer thickness; I—retinal nerve fiber layer thickness in inferior quadrant; S—retinal nerve fiber layer thickness in superior quadrant; N—retinal nerve fiber layer thickness in nasal quadrant; T—retinal nerve fiber layer thickness in temporal quadrant; C/D—cup-to-disk ratio; R/D—rim-to-disk ratio.

## Data Availability

The data presented in this study are available on request from the corresponding author. The data are not publicly available due to General Data Protection Regulation.

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
