# Peer review of "Do Gender, Age, Body Mass and Height Influence Eye Biometrical Properties in Young Adults? A Pilot Study"

_ijerph, 2021, doi:10.3390/ijerph182111719_

Round 1
Reviewer 1 Report
Authors uncovered predictors of ocular biometrics based on data from 155 younger near-emmetropic subjects. This will provide better understanding of ocular morphology and improve biometry-based diagnostic and therapeutic procedures in ophthalmology.
Major revision
- Data of refractive error is lacked in table 1-3. Although authors selected subjects whose refractive error is minimal (-1.0D to +2.50D), biases in refractive error by the group possibly affects the evaluation of axial length and corneal curvature. Therefore, refractive error is better to be included in the evaluation items.
- line 208-,
“We found that body mass had a positive correlation with eyeball axial length (p=0.000; p=0.009) and a negative correlation with the steepest (p=0.001; 0.032) and the flattest (p=0.001; p=0.010) corneal meridian curvature (expressed in diopters) both for male and female subjects. The correlation of body height with eyeball axial length was not significant in female subjects (p=0.000; p=0.086). However, there was a significant correlation for the steepest (p=0.000; 0.007) and the flattest (p=0.000; p=0.005) corneal meridian curvature for both male and female subjects, respectively. Nangia V. et al. studied eye biometry measured by ultrasonography in a rural area of central India in healthy subjects aged 30 years and older. The researchers found that taller subjects had larger eyes with flatter corneas.[8] Taller subjects were more likely to have larger eyes with longer axial length (+0.23 mm for each 10 cm increase in height) and deeper anterior chambers (+0.03 218 mm for each 10 cm increase in height). Using a more precise technique, in a Croatian population, we found that taller males, but not taller females, have longer AL, deeper ACD, flatter corneas, and deeper anterior chambers.”
In this paragraph, authors may confuse body height with body mass. In addition, the content overlaps with the second paragraph.
・The part of this paragraph can be moved to the second paragraph.
・Body height and body may correlate well. Is there an advantage in considering weight as well as height? This needs to be separately discussed.
- lines 268-,
“Considering the normal biometric properties of the eye and their dependence on gender, body height and body mass in younger populations enables the early detection of pathological conditions. Thus, doctors can perform early diagnosis and successfully treat various ocular pathological conditions.
The impact on diagnosis and treatment of ocular disease should not be emphasized in the conclusion section, because the data in this study does not confirm this.
Minor revision
1.
line 39, “To best of our knowledge, this is the first study that has focused on the correlation between the eye biometric properties of near-emmetropic pre- presbyopic healthy adults. Our study includes comprehensive anterior and posterior segment parameters and general biometric parameters, including age, gender, body mass and height.”
Difficult to catch the meaning. English correction required.
- line 168,
Corneal curvature and corneal thickness are important parameters in refractive surgery.
This description can be deleted, because no discussion is found in this paragraph how to use the result of this study in refractive surgery.
3.
line 179,
Other authors found that corneal thickness can be influenced by ethnicity, age, gender and genetics.
Please add reference number. “Corneal thickness is reported to be influenced by ethnicity, age, gender and genetics.”
- line 250,
Patel et al. described the association of RNFL thickness loss with age between 19 and 249 76 years.[29]
However, our data do not show a decrease in the average RNFL thickness with age 251 between 18 and 39 years.
Unnecessary change in paragraphs.
- line 254-,
Our study contributes to a better understanding of biometrical properties, physiological aging processes, and gender and population differences and their relationship to 255 body height and mass.
This can be deleted.
Author Response
Dear reviewer,
thank You for Your comments, please find our responses in attached file.
Sincerely Yours,
Jurica Predović

Reviewer 2 Report
The article deals with a topic of great interest and it could be a basis for further studies.
First of all, I would like to congratulate the authors for addressing such an interesting topic. However, I encourage them to modify some aspects that could improve the research they propose.
Some minor issues to address are:
- The title does not fit well with the article. In my opinion it should not include "background/objectives"
- There are some inconsistencies in the references, which sometimes go behind the point and sometimes not
- Although authors explain how are they calculate the sample, it seem to be a small number of subjects to make so important conclusions. I think you should added in the limitations of the study
Regarding the major changes: considering the amount of data obtained from the subjects, I would take the opportunity to make a more ambitious statistical analysis. I think it would be of great interest to do a multivariate analysis that can explain the weight of each of the parameter studied in the changes in biometrical values.
I encourage the authors to do this multivariate analysis that helps to understand which parameters should be considered to a greater or lesser extent when we take biometric data from different patients.
I hope the can afford these cahnges to make a more valuable research
Author Response

(The authors gave the same response as above.)

Round 2
Reviewer 1 Report
Well revised.
Author Response
Dear reviewer,
thank You for all constructive comments and giving us positive feedback.
Yours sincerely,
Jurica Predović
Reviewer 2 Report
Although authors has answered all my concerns, as far as the multivariate analisys cannot be done in the present study but it is planned for a future one I suggest to change the title from "Do gender, age, body mass and height influence eye biometrical properties in young adults?" to Do gender, age, body mass and height influence eye biometrical properties in young adults? A pilot study"
In adition, I suggest to include in the abstract the backgound of the study to comply with the journal's instructions.
Author Response
Dear reviewer, I agree with You. Thank You for all constructive comments.
My answers are listed below.
Comment 1. Although authors has answered all my concerns, as far as the multivariate analisys cannot be done in the present study but it is planned for a future one I suggest to change the title from "Do gender, age, body mass and height influence eye biometrical properties in young adults?" to Do gender, age, body mass and height influence eye biometrical properties in young adults? A pilot study"
Response 1. I changed the title to " Do gender, age, body mass and height influence eye biometrical properties in young adults? A pilot study "
Comment 2. In adition, I suggest to include in the abstract the backgound of the study to comply with the journal's instructions.
Response 2. Thank you for the suggestion: I included the background of the study in the abstract.
Sincerely Yours,
Jurica Predović